# Clinicopathological Characteristics of Ovarian and Breast Cancer in *PALB2*, *RAD51C,* and *RAD51D* Germline Pathogenic Variant Carriers

**DOI:** 10.3390/genes16050556

**Published:** 2025-05-02

**Authors:** Jella-Rike J. A. H. Spijkervet, L. Lanjouw, L. P. V. Berger, M. D. Dorrius, B. van der Vegt, G. H. de Bock

**Affiliations:** 1Department of Epidemiology, University of Groningen, University Medical Center Groningen, 9713 GZ Groningen, The Netherlands; 2Department of Genetics, University of Groningen, University Medical Center Groningen, 9713 GZ Groningen, The Netherlands; 3Department of Radiology, University of Groningen, University Medical Center Groningen, 9713 GZ Groningen, The Netherlands; 4Department of Pathology & Medical Biology, University of Groningen, University Medical Center Groningen, 9713 GZ Groningen, The Netherlands

**Keywords:** hereditary breast and ovarian cancer syndrome, practice guideline, cancer screening, *PALB2*, *RAD51C*, *RAD51D*, genetic counseling

## Abstract

**Background/Objectives:** Germline pathogenic variants (GPVs) in *PALB2*, *RAD51C*, and *RAD51D* increase breast cancer (BC) and ovarian cancer (OC) risk. Limited data on clinicopathological characteristics of BC and OC in women with these GPVs hamper guideline development. Therefore, this study aims to describe these characteristics in a consecutive series of female *PALB2*, *RAD51C*, and *RAD51D* GPV carriers. **Methods**: Women with a *PALB2*, *RAD51C*, or *RAD51D* GPV determined before July 2023 at the University Medical Center Groningen were included. Cancer diagnoses were obtained through linkage with the Dutch Nationwide Pathology Databank (Palga). Median onset age and histopathological subtypes were compared to the data of The Netherlands Cancer Registry (NCR). **Results**: Among 164 GPV carriers (125 *PALB2*, 30 *RAD51C*, and 9 *RAD51D*), 54 BC and 6 OC cases were identified. The median BC onset age was 52 (*n* = 50), 71 (*n* = 3), and 43 years (*n* = 1) for *PALB2*, *RAD51C*, and *RAD51D*, respectively, compared with 62 years in the NCR. No BC histological subtype differences were observed in *PALB2* carriers. The populations of *RAD51C* and *RAD51D* carriers were too small to compare to NCR data. No OC cases occurred in *PALB2* carriers. The median OC onset age was 66 (*n* = 4) and 56 years (*n* = 2) for *RAD51C* and *RAD51D* carriers, respectively, versus 67 years in the NCR. All *RAD51D* carriers had high-grade serous carcinoma, compared to 51.5% in the NCR. **Conclusions**: Differences in onset age and histological subtypes were observed between GPV carriers and national data. Further research on cancer characteristics is needed to optimize counseling and cancer prevention in these women.

## 1. Introduction

Hereditary factors contribute to approximately 8% of breast cancer (BC) cases and 10–15% of ovarian cancer (OC) cases [1,2]. Germline pathogenic variants (GPVs) in *BRCA1* and *BRCA2*, genes involved in the DNA repair process, are the most common cause of hereditary breast and ovarian cancer (HBOC) [3,4]. In addition, GPVs in the genes *CHEK2, ATM, BARD1, PALB2, RAD51C,* and *RAD51D* are associated with an increased risk of BC [5,6,7,8,9,10]. As with *BRCA1/2*, the latter three genes are also associated with an increased risk of OC, with evidence suggesting that they may lead to an earlier onset of BC and OC compared to the age of onset in the general population [6,9]. Very little is known about the prevalence of GPVs in these genes, but a GPV in the *PALB2* gene is found in approximately 1% of women who are genetically tested [11]. Carriers of a *PALB2* GPV have a relative risk of 7.2 for BC, with an estimated BC risk of 53% by the age of 80 [12]. Previous research has also suggested that BC in *PALB2*, *RAD51C*, and *RAD51D* GPV carriers is more often associated with triple-negative tumors [13,14]. However, to date, there is a lack of comprehensive data describing other tumor characteristics such as histological subtype and tumor size and stage in *PALB2, RAD51C*, and *RAD51D* GPV carriers.

While screening guidelines for *BRCA1/2* GPVs are well defined, recommendations remain unclear for *PALB2, RAD51C*, and *RAD51D* GPVs. Women carrying a *BRCA1/2* GPV can receive intensive breast screening aiming to detect BC at an early stage, starting at age 25 with an annual MRI and consultation. *BRCA1* carriers additionally receive bi-annual mammography from 40 to 60 years, whereas *BRCA2* carriers receive additional yearly mammography between the ages of 30 and 60 [15,16]. In addition, women may undergo preventive surgery of the breast and/or adnexa [15,16]. For the latter, timely preventive surgery is the only effective method to reduce risk, as screening for ovarian cancer is ineffective [17]. Women with *BRCA1* GPVs are advised to undergo this surgery between the ages of 35 and 40, while women with *BRCA2* GPVs are advised to do so between the ages of 40 and 45 [18]. In contrast, data on cancer risk for *PALB2*, *RAD51C*, and *RAD51D* GPV carriers are limited and lifetime cancer risks largely depend on both family history and personal risk factors, such as age of menarche, use of oral contraceptives, and the number of children. Dutch guidelines regarding BC surveillance in women with a *PALB2* GPV currently recommend yearly MRIs, mammograms, and consultations between the ages of 30 and 60, which are limited to mammograms and consultations from the age of 60 to 75 [19]. Advice for BC screening for *RAD51C* and *RAD51D* GPV carriers depends on personal lifetime BC risk estimations calculated by the Canrisk tool [5,20,21,22]. This tool is used to assess the lifetime OC risk in *PALB2, RAD51C,* and *RAD51D* GPV carriers as well, leading to personalized advice on risk-reducing salpingo-oophorectomy (RRSO). There is no clear recommendation on the timing of an RRSO in these women.

The limited knowledge of BC and OC risk, age of onset, and associated histological subtypes with these GPVs, complicates evidence-based counseling and preventive care for women with a *PALB2, RAD51C,* or *RAD51D* GPV. Therefore, this study aims to describe the age of onset and clinicopathological characteristics of BC and OC patients with a *PALB2, RAD51C,* or *RAD51D* GPV and compare these characteristics to national cancer registry data.

## 2. Materials and Methods

### 2.1. Setting

According to Dutch guidelines, women diagnosed with BC at a young age or those with risk factors, such as a family history of BC or a Jewish background, are eligible for germline testing [15]. At the University Medical Center Groningen (UMCG), this is performed by the Department of Genetics, in line with Dutch guidelines. The panel currently applied includes the *BRCA1, BRCA2, PALB2, CHEK2, ATM, RAD51C, RAD51D*, and *BARD1* genes. Since 2015, all women with OC have been eligible for germline testing, regardless of risk factors [18]. In 2022, national guidelines changed and now recommend a tumor-first approach for all OC patients, in which pathogenic variants (PV) are tested in tumor tissue. Patients with a PV in the tumor, and those with a family history of OC, are referred for a germline test. The genes analyzed by the Department of Genetics of the UMCG are *BRCA1, BRCA2, BRIP2, PALB2, RAD51C, RAD51D, MLH1, MSH2, EPCAM*, and *MSH6*. If a hereditary GPV is detected, relatives are eligible for counseling and genetic testing as well. Further counseling is carried out by a team of specialists, including, among others, a clinical geneticist, a surgical oncologist, a psychologist, and a gynecological oncologist, in the breast–ovarian carcinoma workgroup specializing in hereditary BC and OC cases.

### 2.2. Study Population

This cohort study included a consecutive series of women (≥18 years) who tested positive for a *PALB2, RAD51C*, or *RAD51D* GPV before 15 July 2023 at the UMCG. Women were eligible for genetic testing due to (1) a previous BC or OC diagnosis (in a close relative), or (2) a family member carrying a *PALB2, RAD51C,* or *RAD51D* GPV. Women were tested for these GPVs using one of the above-mentioned panels or by testing specifically for the familial *PALB2, RAD51C,* or *RAD51D* GPV if applicable. Women with a variant of unknown significance (VUS) in one of these genes were excluded as pathogenicity was not established. Also, clinical guidelines are applicable to GPV carriers and not VUS carriers in *PALB2, RAD51C*, or *RAD51D* GPVs. The VUS was not reassessed over the study period. Following Dutch law, the study protocol was proposed to the Medical Ethics Review Board (METc) of the UMCG to review whether human subjects, as decided in the WMO act, were involved. The METc UMCG declared this a non-WMO study as the research does not involve human objects as described in the WMO (Metc 2023/136).

### 2.3. Study Design

This was an observational, retrospective cohort study. The characteristics of OC and BC tumors of women with a *PALB2*, *RAD51C*, or *RAD51D* GPV were compared with publicly available national cancer data from The Netherlands Cancer Registry (NCR) [23].

### 2.4. Data Collection

In this study, we collected data on GPV status, date of testing, date of birth (month and year), date of diagnosis of cancer (month and year), histological subtype of cancer, hormone receptor (HR) status, tumor grade, and tumor size at time of diagnosis. Type of GPV, date of genetic test result, and date of birth were collected from the Department of Genetics of the UMCG. Pathology reports from women with a GPV were obtained through linkage with the Dutch Nationwide Pathology Databank (Palga), which provides full coverage of histopathological reports [24]. From the Palga reports, the date of cancer diagnosis, type, histological subtype, HR status, tumor grade, and tumor size at the time of diagnosis were extracted. This research primarily focused on primary breast or ovarian malignancies. For women with multiple cancer diagnoses, the age of onset was determined based on the first occurrence of BC or OC, even if a different type of cancer was diagnosed at an earlier age. In cases where both a breast and ovarian tumor were present, distinct ages of onset were calculated for each malignancy. The tumor size was analyzed at the time of diagnosis. National cancer data were obtained from the NCR and included data about all cancer diagnoses and patient characteristics in The Netherlands. The median age of onset and incidence of the different histological subtypes of BC and OC over the period 2018–2022 were obtained [23].

### 2.5. Variable Definition and Statistical Analysis

The age of onset of BC or OC was calculated as the number of years between the date of birth (first day of the month of birth) and the date of primary cancer diagnosis (first day of the month of diagnosis). The age of onset was reported as the median, and the interquartile range (IQR) was reported where possible. When the number of cases was insufficient to calculate the IQR, this was reported as not applicable (n.a.). The cancer type and pathohistological subtype were reported as frequencies and percentages. The tumor diameter at the time of diagnosis was reported as the median and IQR. The age of BC and OC onset was compared with national data from the NCR using frequencies and percentages.

## 3. Results

### 3.1. Patient Characteristics

A total of 164 women were included. The median age at the time of genetic testing was 53.3 (IQR: 40.0–65.0) years (Table 1). There were 125 (76.2%) women with a *PALB2* GPV, 30 (18.3%) women with a *RAD51C* GPV, and 9 (5.5%) women with a *RAD51D* GPV. In *PALB2* GPV carriers, co-occurrence of a GPV in the *CHEK2* gene and *NF1* gene was found in four and two cases, respectively. Co-occurrence of a GPV in *BRCA1* was present in one *RAD51C* GPV carrier. A primary tumor was diagnosed in 76 (46.3%) women (Figure 1). As one woman developed both OC and BC, there were 77 primary tumors. Out of the 76 women who developed cancer, there were 54 (71.1%) who developed BC and 6 (7.9%) who developed OC. Seventeen (22.4%) women were diagnosed with another type of primary cancer (Appendix A). Out of 54 BC patients, 11 (20.4%) women experienced a recurrence of BC. One woman had a recurrence of OC. Of the 76 women with a primary tumor, 65 (85.5%) women were genetically tested after having their cancer diagnosis. Nine (11.8%) women were genetically tested before having their first cancer diagnosis, and for two women, the date of cancer diagnosis was unknown.

### 3.2. Breast Cancer

Of the 54 women diagnosed with BC, 50 women carried a *PALB2* GPV. Three women carried a *RAD51C* GPV, and one woman carried a *RAD51D* GPV (Table 2). In women diagnosed with BC, co-occurrence of a *CHEK2* GPV was present in one woman carrying a *PALB2* GPV. No further co-occurrence of GPVs was present in women diagnosed with BC. The median age of onset of BC in women with a *PALB2* GPV was 52.0 (IQR: 42.0–61.5) years. In women with a *RAD51C* GPV, this was 71.0 years, and for the woman with a *RAD51D* GPV, this was 43.0 years. The median age of BC onset in the data of the NCR is 62.0 (IQR: 51.0–72.0) years. Of the 50 BC patients with a *PALB2* GPV, 45 (90.0%) had an invasive carcinoma of no special type (NST). Invasive lobular carcinoma was diagnosed in four (8.0%) women, and one woman was diagnosed with another subtype. Among the *RAD51C* carriers, one woman was diagnosed with invasive carcinoma NST, one woman with lobular carcinoma, and one woman with another subtype. The woman with a *RAD51D* GPV had an invasive lobular carcinoma. In the NCR data, 79.7% of the BCs are invasive carcinoma NST, while 13.0% are invasive lobular carcinoma. A tumor grade was reported for 39 (78.0%) *PALB2* GPV carriers; grade 1 five times (12.8%), grade 2 twenty-one times (53.8%), and grade 3 thirteen times (33.3%). In two *RAD51C* GPV carriers, the tumor grade was reported, with both being grade 1. In the woman carrying a *RAD51D* GPV, tumor grade 2 was found. The tumor diameter was reported in 31 (62.0%) *PALB2* carriers. The median tumor diameter at the time of diagnosis was 15.0 mm (IQR: 11.0–18.0). The tumor diameter at the time of diagnosis was reported in two (66.7%) *RAD51C* carriers, with a median of 20.5 mm (IQR: n.a.). The tumor diameter of the patient carrying a *RAD51D* GPV was not reported. Out of all 50 BC patients carrying a *PALB2* GPV, HR status was reported for 44 (88.0%) patients, and HER2 status for 39 (78.0%) patients. Out of the 44 patients, 32 (72.7%) had an HR-positive tumor and 3 (7.7%) out of 39 women had a HER2 positive tumor. Out of all 50 BC patients, nine (18.0%) had a triple-negative tumor, and four (8.0%) patients did not have any information on hormone receptor status. All three (100%) patients carrying a *RAD51C* GPV had an HR-positive and HER2-negative tumor. The *RAD51D* carrier had a triple-negative tumor. Within the NCR data, 81.7% of the tumors are HR positive, 13.0% HER2 positive, and 11.1% triple negative.

### 3.3. Ovarian Cancer

None of the *PALB2* GPV carriers were diagnosed with OC (Table 2). Out of the six women diagnosed with OC, four carried a *RAD51C* GPV, and two carried a *RAD51D* GPV. The median age of onset of OC in the four women with a *RAD51C* GPV was 66.0 (IQR: 61.0–69.5) years. In the two women with a *RAD51D* GPV, this was 56.0 years. The median age of OC onset in the NCR data is 67.0 (IQR: 59.0–76.0). Of the four women with a *RAD51C* GPV, three were diagnosed with a high-grade serous carcinoma. In the *RAD51D* GPV group, both patients were diagnosed with high-grade serous carcinoma. In the NCR data, 51.5% of the OCs are classified as high-grade serous carcinoma.

## 4. Discussion

This cohort aimed to describe clinicopathological characteristics in a consecutive series of female *PALB2*, *RAD51C*, or *RAD51D* GPV carriers. In this cohort, the median age of BC onset was lower for *PALB2* and *RAD51D* GPV carriers but higher for *RAD51C* GPV carriers compared to the data of the NCR. OC was found only in *RAD51C* and *RAD51D* GPV carriers, with a lower median age of onset compared to the NCR data. No substantial differences were observed in the age of onset or distribution of OC histological subtypes between women with a *PALB2* or *RAD51C* GPV and the NCR data.

This study reports a lower age of BC onset in *PALB2* GPV carriers compared to the NCR data; 52 years compared to 62 years. This is consistent with previous research reporting that more than 50% of *PALB2* carriers were diagnosed with BC before the age of 50 years, with an eight to ninefold increased risk under the age of 40 [25]. Another study found a median age of BC onset of 40.25 years (range: 25–53) in *PALB2* carriers [26]. Our study observed that the distribution of histological BC subtypes in *PALB2* GPV carriers was similar to the distribution in the NCR data. Li et al. found that 75% of all BC cases in *PALB2* GPV carriers were of the invasive carcinoma NST subtype, which is lower than our finding of 90.2% [27]. Our study showed that 18.0% of *PALB2* carriers had a triple-negative tumor, which is higher than the incidence reported in the NCR data (11.1%). This is in line with a study that found a higher frequency of triple-negative tumors in *PALB2* carriers [28].

A large study on BC and OC in *RAD51C* and *RAD51D* carriers reported an increased risk of developing BC and/or OC in *RAD51C* (*n* = 215) and *RAD51D* (*n* = 92) GPV carriers [5]. That study also found an increased risk of early-onset BC and OC, with over half of the BC patients with a *RAD51C* or *RAD51D* GPV being under the age of 50 [5]. In contrast, our study found a median BC onset age of 71 years in *RAD51C* GPV carriers, with the earliest case at 47 years. This could be the result of our small study population. Our finding of a BC onset age of 43 in a *RAD51D* GPV carrier is consistent with the study of Yang et al., as they found that the BC onset age in *RAD51D* GPV carriers (*n* = 30) was below the age of 50 years in over 60% of cases, with over 33% being between the ages of 40 and 49 [5]. A study on OC in *RAD51C* and *RAD51D* carriers found that the majority of patients with OC and a *RAD51C* GPV were above the age of 50 years [29]. Our study showed similar results; a median age of OC onset in *RAD51C* GPV carriers of 66 years with all cases occurring between 60 and 70. For *RAD51D* GPV carriers, one study found that more than 75% of OC cases were between the ages 50 and 69 [5], while others found that more than 90% of *RAD51D* GPV carriers with OC were above the age of 50 [28]. This aligns with our findings of a median OC onset age of 56 in *RAD51D* GPV carriers. We have found no additional studies on the age of BC and OC onset, or on histological subtypes, in *RAD51C* or *RAD51D* GPV carriers. The literature does show that OC in women with a *BRCA1* or *BRCA2* GPV is generally associated with high-grade serous ovarian carcinoma [30]. We found 75% and 100% of OC cases of the high-grade serous subtype in women with a *RAD51C* and *RAD51D* GPV, respectively. *RAD51C* and *RAD51D* genes are, similar to *BRCA1* and *BRCA2*, involved in homologous recombination repair pathways, possibly implying that there is an association between these GPVs and high-grade serous OC as well.

### Strengths and Limitations

A strong point of this study is the inclusion of a consecutive series of women carrying a *PALB2, RAD51C*, or *RAD51D* GPV who were all genetically tested in a specialized center with a strict protocol. A second strength of this study is the linkage with the pathology database, Palga, as this ensured 100% coverage of pathological reports in The Netherlands. This way, patients diagnosed with cancer outside our institution were also included. Also, analyzing the tumor diameter, tumor grade, histological subtype, and receptor status in BC patients led to a complete overview of the histopathological characteristics of these groups. Lastly, to our knowledge, this is the first study investigating the histological subtype of OC in *RAD51C* and *RAD51D* GPV carriers. However, several limitations must also be considered. First, like most studies on *RAD51C* and *RAD51D* GPVs, the number of women included was relatively low. Subgroup and association analysis could not be performed due to this small sample size, which limited the ability to draw conclusions on differences between these groups. Future studies, preferably multi-center, may overcome this limitation by including a larger cohort. Secondly, the testing and screening protocols determined which women were identified and followed, thereby potentially influencing the observed age of diagnosis through detection bias. Although our cohort represents a clinically relevant population, it may have resulted in a younger observed age of onset compared to what might have been seen in an unselected group of *PALB2, RAD51C,* or *RAD51D* GPV carriers.

In this consecutive series, a lower age of BC onset was seen in women with a *PALB2* GPV compared to the NCR data. In women with a *RAD51C* GPV, a higher age of BC onset and a lower age of OC onset were observed compared to the NCR data. A lower age of onset was observed for both BC and OC in women with a *RAD51D* GPV compared to the NCR data. Additional studies with a larger sample size, for example, by pooling existing cohorts, are required to study the histological subtypes and age of onset in women with a *PALB2*, *RAD51C*, or *RAD51D* GPV to draw more reliable conclusions.

## Figures and Tables

**Figure 1 genes-16-00556-f001:**
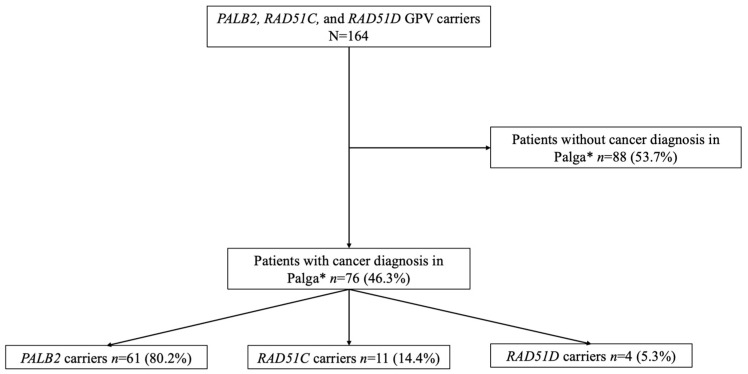
Flowchart of Palga linkage results. * Patients diagnosed with cancer in The Netherlands can be linked to Palga; without a cancer diagnosis, there is no linkage with Palga.

**Table 1 genes-16-00556-t001:** Characteristics of female GPV carriers and their tumors (*n* = 164) *.

Characteristics	Total N = 164
Age at the time of genetic testing in years (median (IQR))	54.0 (40–65)
Type of mutation	
*PALB2*	125 (76.2%)
*RAD51C*	30 (18.3%)
*RAD51D*	9 (5.5%)
Women with a diagnosis of primary cancer **	76 (%)
*Breast*	53 (69.7%)
*Ovarian*	5 (6.6%)
*Breast and ovarian*	1 (1.3%)
*Other*	17 (22.4%)
Second diagnosis/Recurrence	12
*Breast*	11
*Ovarian*	1
Sequence of genetic testing and cancer diagnosis	76 (%)
*Cancer diagnosed before genetic testing*	65 (85.5%)
*Cancer diagnosed after genetic testing*	9 (11.8%)
*Unknown* ***	2 (2.6%)

* Presented are *n* (%) unless specified otherwise; ** one patient had both primary OC and BC diagnosis; *** for two patients, the exact date of cancer diagnosis was unknown.

**Table 2 genes-16-00556-t002:** Characteristics of breast and ovarian cancer within GPV groups.

	*PALB2*	*RAD51C*	*RAD51D*	NCR Data
**Breast cancer (*n*)**	50	3 *	1	
Age at the time of diagnosis (median (IQR))	52 (42–61.5)	71 (n.a.) **	43 (n.a.) **	62 (51–72)
Histological subtype				
*Invasive carcinoma* *NST*	45 (90.0%)	1 (33.3%)	0	79.7%
*Invasive lobular* *carcinoma*	4 (8.0%)	1 (33.3%)	1 (100%)	13.0%
*Other subtypes*	1 (2.0%)	1 (33.3%)	0	7.3%
Tumor grade				
*1*	5 (12.8%)	2 (100%)	0	-
*2*	21 (53.8%)	0	0	-
*3*	13 (33.3%)	0	1 (100%)	-
*Not reported*	11 (22.0%)	1 (33.3%)	0	-
Tumor diameter at the time of diagnosis (median (IQR))	15.0 mm (11.0–18.0)	20.5 mm (n.a.) **	Unknown	-
Hormone receptor status				
*HR +*	32 (72.7%)	3 (100%)	0	81.7%
*HER2 +*	3 (7.7%)	0	0	13.0%
*Triple negative*	9 (18.0%)	0	1 (100%)	11.1%
*Unknown*	4 (8.0%)	0	0	
**Ovarian cancer (*n*)**	0	4 *	2	
Age at the time of diagnosis (median (IQR))	-	66 (61–69.5)	56 (n.a.) **	67 (59–76)
Histological subtype				
*High-grade serous*	-	3 (75.0%)	2 (100%)	51.5%
*Low-grade serous*	-	-	-	4.9%
*Endometroid*	-	-	-	5.9%
*Clear cell*	-	-	-	5.2%
*Mucinous*	-	-	-	5.7%
*Carcinosarcoma*	-	-	-	2.0%
*Other*	-	1 (25.0%)	-	24.7%

Bold font was used to differentiate between breast and ovarian cancer patients with PALB2, RAD51C and RAD51D germline pathogenic variants. * One patient had both BC and OC as the primary tumor. ** n.a.: not applicable; cannot be calculated due to small number of patients.

## Data Availability

The original contributions presented in the study are included in the article/Appendix A, further inquiries can be directed to the corresponding authors.

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
