# Peer review of "Clinicopathological Characteristics of Ovarian and Breast Cancer in PALB2, RAD51C, and RAD51D Germline Pathogenic Variant Carriers"

_genes, 2025, doi:10.3390/genes16050556_

Round 1
Reviewer 1 Report
Comments and Suggestions for Authors
I would like to commend the authors on this excellent manuscript. The study addresses a significant gap in the literature regarding the clinicopathological characteristics of breast and ovarian cancer in women carrying germline pathogenic variants (GPVs) in PALB2, RAD51C, and RAD51D. The methodology is sound, the linkage with the Dutch Nationwide Pathology Databank (Palga) is a notable strength, and the comparisons with the Netherlands Cancer Registry (NCR) provide valuable context for interpretation.
The manuscript is well-written, clearly structured, and presents its findings in a balanced and informative manner. Despite the relatively small sample sizes for RAD51C and RAD51D carriers, the authors are appropriately cautious in their interpretations and highlight the need for further research. The clinical implications, especially for personalized counseling and preventive strategies, are clearly articulated and underscore the relevance of this work.
I have no major concerns and only minor editorial suggestions that can be addressed during copyediting.
Reviewer 2 Report
Comments and Suggestions for Authors
While PALB2, RAD51C, and RAD51D germline pathogenic variants (GPVs) have increasingly been recognized as contributors to hereditary breast and ovarian cancer (BC/OC), comprehensive data on their associated cancer risk and specific clinicopathological features remain limited. In this study, the analysis of these genes in terms of penetrance, phenotype correlations, and potential interactions with BC/OC was insufficiently addressed.
1. Introduction
“In contrast, data on cancer risk for PALB2, RAD51C and RAD51D GPV carriers are limited...”
This introduction lacks adequate context about the previously known characteristics of PALB2, RAD51C, and RAD51D GPVs in relation to breast and ovarian cancer. Even though the available literature may be limited, it would be valuable to provide a concise overview of the known or hypothesized cancer risk associated with these GPVs. Clarifying whether these genes have only been previously linked with early-onset cancers would provide essential background and help frame the study’s significance.
- Materials and Methods
Section 2.2: Study Population
“Women with a variant of unknown significance (VUS) in one of these genes were excluded.”
The rationale behind excluding VUS (variants of unknown significance) in PALB2, RAD51C, and RAD51D is not clearly justified. Given the limited data on these genes and their evolving classification, VUS may still hold exploratory value in such a study. Including VUS (at least in a secondary analysis or with annotation) could potentially help uncover emerging risk patterns, especially when considering broader clinical and pathological criteria. A brief explanation addressing this exclusion — and whether these VUS were reassessed during the study period — would add clarity.
- Results 1)Descriptive & Lack of Association Analysis
The current results are mostly descriptive and summarize the cohort’s characteristics without conducting more robust association analyses. The study would benefit from deeper investigation into the relationship between specific clinicopathological features and individual GPVs.
Suggestions: conduct association analysis
- Is a specific PALB2 GPV linked with invasive carcinoma or specific tumor grades?
- Are there histologic or molecular patterns (e.g., triple-negative BC) enriched in certain gene carriers?
- ...
2) Confounding Genetic Factors: BRCA1/2 and Others
It is critical to evaluate the co-occurrence of BRCA1 and BRCA2 mutations in patients carrying PALB2, RAD51C, or RAD51D GPVs. Given that BRCA1/2 mutations are strongly associated with BC/OC, their presence could confound or modify the observed effects of the study genes.
Suggestions:
- Subgroup analysis: stratify patients as +/− BRCA1/2.
- Compare the clinicopathological characteristics between isolated PALB2/RAD51C/RAD51D GPVs vs. co-mutations with BRCA1/2.
- Report mutation rates of other known cancer-associated genes (if multigene panel testing was done).
